# The Quiet Epidemic: An Overview of Emerging Qualitative Research Trends on Sedentary Behavior in Aging Populations

**DOI:** 10.3390/healthcare11152215

**Published:** 2023-08-06

**Authors:** André Ramalho, João Petrica

**Affiliations:** Sport, Health & Exercise Research Unit (SHERU), Polytechnic Institute of Castelo Branco, 6000-266 Castelo Branco, Portugal; j.petrica@ipcb.pt

**Keywords:** sedentary behavior, older adults, qualitative research, interventions

## Abstract

Sedentary behavior poses a significant health risk to older adults. The purpose of this scoping review is to summarize key findings from qualitative research (since 2012), with a focus on explaining prevailing research trends and patterns within the field. This review included 25 studies that met the strict inclusion criteria. Five key research themes have emerged: (1) defining and shaping perspectives of sedentary behavior, (2) understanding the dynamics of daily routines and contexts, (3) raising awareness of older adults’ perceived advantages and disadvantages of sedentary behavior, (4) identifying its determinants and discouraging factors, and (5) exploring interventions to promote active behaviors and reduce sedentary behavior. These themes highlight the multifaceted nature of SB and underscore the importance of tailoring interventions to address individual, social, and environmental issues. A comprehensive understanding of SB is critical to developing effective strategies to promote active lifestyles and reduce SB in older adults. Further qualitative research is needed to deepen our understanding and develop targeted interventions and strategies.

## 1. Breaking Ground on Sedentary Behavior: An Introduction to a Silent Threat

Sedentary behavior (SB) is known to be an essential part of the daily routine of older adults, exceeding eight hours per day [1,2,3]. The concept is defined as any waking behavior that involves minimal energy expenditure (less than 1.5 metabolic equivalents) while sitting, leaning, or lying down [4]. Higher levels of daily SB have been associated with significant health outcomes for adults, particularly older adults [5]. Regardless of physical activity level, prolonged SB is associated with poor health outcomes, including obesity, diabetes, heart disease, cancer, and premature mortality [6,7,8]. These findings highlight the detrimental effects of a sedentary lifestyle on overall health and emphasize the importance of promoting physical activity and reducing sedentary time to mitigate these risks. Importantly, these adverse consequences persist regardless of a person’s level of physical activity, underscoring the independent effects of prolonged SB on biopsychosocial health [6]. Therefore, SB goes beyond simply abstaining from moderate to vigorous physical activity. It can be effectively reduced by incorporating more transitions from sitting to standing, increasing standing time, and engaging in light physical activities, including active sitting [9].

SB includes both passive activities (e.g., watching television) and intellectually active behaviors (e.g., reading or using a computer). These activities occur in a variety of behavioral contexts, such as work, leisure, and transportation [10,11,12]. Research with older adults suggests that certain sedentary activities, such as reading or working on a computer, can improve cognitive performance and promote mental well-being [13]. Intellectually stimulating tasks, for example, may contribute to better mood, lower stress levels, and improved emotional well-being [14,15]. Therefore, it is important to recognize that not all sedentary activities have uniformly negative effects. Rather, the effects are influenced by the context and type of activity. While excessive SB remains a health concern for older adults, it is important to recognize the potential benefits and differential effects of certain sedentary activities.

Research on SB in older adults has increased in recent years [16]. Literature reviews have used the behavioral epidemiology framework to gain a comprehensive understanding of SB in this population [17]. Numerous health conditions, including obesity, cardiovascular disease, and type 2 diabetes, have been consistently associated with SB [18,19,20,21]. Studies have focused on quantifying and measuring the variability of SB in older adults, examining factors such as daily sitting duration or patterns of SB [22,23,24,25]. Determinants of SB have also been studied in depth, highlighting factors such as age-related decline and lack of awareness of the negative health effects of prolonged sitting [26,27,28]. In addition, interventions to address daily SB have been developed and evaluated, and their effectiveness has been summarized in comprehensive reviews. Strategies such as standing breaks, community-based programs, and reminders to interrupting prolonged sitting have shown promise [29,30,31]. Overall, these studies provide valuable insight into the effects of SB and suggest potential avenues for interventions aimed at reducing this behavior in older adults.

Qualitative research can develop robust strategies and interventions because of its many advantages. These advantages include a deep understanding of complex phenomena, comprehensive exploration of contextual subtleties that permeate health-related behaviors and interventions, meaningful engagement of stakeholders in the research endeavor, use of an iterative methodology that promotes continuous improvement, and the ability to complement quantitative data with deep insights and broad reach. Currently, however, there is a significant gap in consolidating key themes emerging from qualitative studies of SB among older adults, which limits our understanding of their interpretations, experiences, and attitudes toward SB. Bridging this gap is critical to synthesizing data from multiple qualitative studies and gaining a comprehensive understanding of older adults’ perspectives. This, in turn, can facilitate the development of effective interventions to reduce SB and promote healthy aging. By identifying common themes, patterns, and knowledge gaps, researchers can gain valuable insights and more accurately define research problems [32]. A thorough review of the relevant literature is essential to provide an overview of scientific advances in the study of SB in the older population. It is important to assess the past, present, and future of research in this area to provide information for future studies and interventions [5,13].

Examining the state of scientific knowledge provides valuable insight into the progress and methodology in a particular area of research [33]. By analyzing the existing literature, we can gain a comprehensive picture of the current state of research, identify trends and gaps, and recognize progress made over time. This process facilitates a thorough assessment of the knowledge base, provides information for future research directions, and contributes to the overall progress of the field. The purpose of this scoping review, therefore, is to summarize key findings from qualitative research (since 2012), with a focus on the prevailing trends and patterns of SB among older adults. By analyzing these qualitative studies, we aim to illuminate the major themes, patterns, and findings that have emerged. This review will serve as a valuable resource for researchers, practitioners, and policymakers by providing an overview of the qualitative studies to date. It will help identify gaps in knowledge, areas that need further exploration, and potential avenues for future research and practical implications.

## 2. Crafting the Map: Eligibility Criteria, Search Strategy, Study Selection, Data Extraction and Analysis

This review was carefully conducted according to the guidelines of the scoping review approach [34], which is recognized as an indispensable tool in the ever-growing repertoire of evidence synthesis approaches. The criteria for selecting and excluding primary studies were guided by the acronym SPIDER (Sample, Phenomenon of Interest, Design, Evaluation, Research type) [35].

Sample. The synthesis included primary studies that examined the perceptions of different groups of older people, aged 60 years or older (or studies with a mean age of ≥60 years), living in their own homes or in nursing homes/facilities, with or without multiple comorbidities, and who did or did not participate in guided physical activity programs or who did or did not engage in regular physical activity without guidance. The synthesis also included studies that focused on clinical populations (e.g., older adults undergoing cardiac rehabilitation) and non-clinical populations.

Phenomenon of interest. Qualitative studies of SB perceptions in the older population.

Design. To meet the purposes of this review, our analysis focused on primary studies that used qualitative methodological approaches. We considered studies that used qualitative data collection methods (e.g., unstructured interviews, semi-structured interviews, structured interviews, focus groups, and direct observation) and various qualitative approaches to data analysis (e.g., thematic analysis, interpretive phenomenological analysis, and content analysis).

Evaluation. Qualitative analysis of experiences, feelings, views, and opinions of older adults.

Research type. Studies from all countries published in peer-reviewed English-language journals since 2012. To meet the purposes of this review, our analysis focused on primary studies that used qualitative methodological approaches. We considered studies that used qualitative data collection methods (e.g., unstructured interviews, semi-structured interviews, structured interviews, focus groups, and direct observation) and various qualitative approaches to data analysis (e.g., thematic analysis, interpretive phenomenological analysis, and content analysis). However, we excluded primary studies that collected data using qualitative methods but did not include qualitative analysis (e.g., surveys analyzed using descriptive statistical methods). In addition, conference proceedings, book chapters, and unpublished manuscripts were not included in the evidence synthesis.

A search strategy (between January and April 2023) was developed to search the literature in multiple electronic scientific databases, including PubMed, Web of Science, and Scopus, which were selected for their potential to identify relevant studies related to the research purpose. Primary studies were searched using specific keywords linked to groups of search terms, such as SB and related terms (e.g., sedentary lifestyle, prolonged sitting, sitting time, reclining time, computer time, internet time, television time, screen time, reading time, computer play, and transportation time); qualitative research designs and analyses (e.g., ethnography, phenomenology, life stories, basic theory, case studies, focus groups, descriptive qualitative studies, qualitative analysis, thematic analysis, content analysis, and interpretive phenomenological analysis); and participant characteristics (e.g., aging, senior, older adult, older, and geriatric). The different keywords within each group were combined using the Boolean operator, “OR”, and further combined using the Boolean operator, “AND”. In addition, the reference lists of primary studies selected for synthesis were reviewed to identify new studies that met the defined eligibility criteria.

The search strategy for the PubMed, Web of Science, and Scopus scientific databases (Table 1) was used as the basis for the literature search.

The titles and abstracts of primary studies obtained through the thorough literature search described above were systematically extracted and duplicates were excluded. Subsequently, the authors of this review independently assessed the titles and abstracts of each study against the stated eligibility criteria to decide whether to include or exclude them. In the event of disagreement between the authors on the inclusion or exclusion of studies in the evidence synthesis, agreement was reached through constructive discourse. The authors then analyzed the full texts of the selected studies from the previous phase. This careful review allowed the authors to ensure that the primary studies met the established eligibility criteria. Any disagreements that arose regarding the inclusion or exclusion of studies in the final synthesis were resolved expeditiously and amicably.

The included studies were systematically collected (data extraction) for their characteristics: authors, year of publication, country, research purposes, study population, data collection and data analysis methods, and main topics to provide a comprehensive overview of qualitative research in the field. These details are summarized in Table 2. In addition, this review used a thematic analysis approach [36], which involved a process of reviewing and interpreting the data to identify recurring patterns and themes. Researchers first became familiar with the data and assigned initial codes to categorize the information. Data relevant to each code were then grouped into clusters of related content. In an iterative process, these codes were refined to identify broader themes that captured the essence of the data set. Themes were validated by comparison to the coded excerpts to ensure coherence and representativeness. This rigorous approach provided a solid framework for analyzing the qualitative data and enabled meaningful results.

## 3. From Palette to Canvas: Study Characteristics

The flowchart shown in Figure 1 provides an insightful overview of the results of the literature search. During the scoping review, a total of 2832 entries were initially identified through the database search, and two additional studies were found through the reference list. After duplicates were filtered out, the number was reduced to 641 studies. From this pool, a careful screening process excluded 605 articles with irrelevant titles and abstracts, leaving 36 full-text articles for further analysis. Finally, 11 of these articles were excluded after a full-text reading; the reasons for their exclusion are carefully explained in Figure 1. Ultimately, 25 studies were selected that met the strict inclusion criteria and provided a compelling rationale for their inclusion in this reexamination. The rigorous selection process ensured that only relevant studies consistent with the research purposes were included in this review.

Table 2 provides an overview of the main characteristics of the 25 studies included in the analysis. The studies were conducted in different countries: eight in the UK, seven in the USA, six in Canada, and one each in Brazil, India, and Sweden. All studies used a qualitative research design. Sample sizes ranged from 9 to 46 participants, and the average age of participants ranged from 63.0 to 83.5 years. Most studies focused on healthy, community-dwelling older adults, while others examined special populations such as patients undergoing cardiac rehabilitation, stroke survivors, older adults with dementia, obese older adults, prostate cancer survivors, and older adults with knee osteoarthritis/total knee arthroplasty. Data collection methods included semi-structured interviews, focus groups, a combination of interviews and focus groups, and workshops. For data analysis, thematic analysis, content analysis, and framework analysis were used in different studies.

## 4. Cracking the Code of Sedentary Behavior: Key Themes Explored and Discussed

### 4.1. Definition and Shaping Perspectives on Sedentary Behavior

SB has significant public health implications, as noted above, particularly among older adults who are more susceptible to chronic disease. However, the precise definition of SB varies, presenting a challenge in developing effective interventions. This challenge is also underscored by qualitative studies that have examined older adults’ views of SB, revealing its nuanced and multifaceted nature. These studies emphasize the complexity of SB and the need for a comprehensive approach to address this phenomenon.

Older adults show some understanding of SB, although they have difficulty distinguishing it from physical inactivity (i.e., not following recommended physical activity guidelines). Their descriptions of SB, such as “sitting around”, “chilling”, or “putting their feet up”, reflect this perception. On the other hand, they use terms like “mooching”, “brooding”, or “hanging out” to describe non-sedentary light activity. In their understanding, SB is often equated with a lack of physical activity or exercise, with sedentary and active behaviors seen as opposite ends of a linear continuum. Consequently, older adults find it difficult to imagine people engaging in high levels of physical activity while sitting for extended periods of time [9]. Similar results have been observed in other populations, including prostate cancer survivors receiving androgen deprivation therapy [56] and individuals with osteoarthritis and knee arthroplasty [59].

When older adults are asked about strategies to reduce SB, they often suggest increasing physical activity instead, and they have difficulty making alternative suggestions that do not involve physical activity unless specifically asked [9]. This confusion may be due to the negative framing of messages that focus primarily on reducing SB. While older adults are able to provide specific examples of SB, such as arts and crafts, the use of technology, gaming, and socializing, the concept of reducing SB is often perceived as ambiguous [49,59]. In addition, older adults desire concrete examples and practical guidance on how to effectively reduce SB, suggesting that clear and specific recommendations would be beneficial [9].

For older adults, distinguishing between SB and physical inactivity can be challenging because these behaviors overlap and there are no precise definitions or clear boundaries. This can lead to confusion [57,59]. SB refers to activities performed while sitting or lying down with low energy expenditure [4], whereas physical inactivity refers to a lack of moderate to vigorous physical activity [61]. It is important to note that a person can be sedentary and physically active at the same time. In addition, societal and cultural norms influence how older adults perceive and understand SB and physical inactivity. Prevailing societal beliefs and stereotypes about aging often normalize SB in older adults [39,59]. This normalization, combined with a focus on physical activity as an indicator of health, can lead to confusion and hinder older adults’ ability to distinguish between SB and physical inactivity. The fact that older adults know little about the difference between SB and physical inactivity underscores the need for education to promote a better understanding of SB and its health implications.

The collective findings of the studies offer insight into the widespread negative perceptions associated with SB among older adults [9,39,44,49,58], using terms such as “doing nothing” and “not moving” to describe it [49]. Older adults report several disadvantages of sitting, including physical discomfort such as stiffness and pain, as well as feelings of boredom and a decline in overall health. They tend to associate SB with the inevitable consequences of aging and with physical limitations [44,58]. Older adults also believe that prolonged sitting negatively affects their mental well-being and leads to more worry and anxiety [44]. In this population, prolonged SB, especially excessive television viewing, can cause guilt, depression, boredom, and idleness [52,58]. Older adults also tend to judge their peers who lead sedentary lifestyles, describing them with derogatory terms such as “stick in the muds” and “cabbages”. Conversely, they often express sympathy for people who lead highly sedentary lifestyles [9].

Older adults are often reluctant to identify themselves as sedentary because of the negative connotations and a desire to distance themselves from this label. They prefer to emphasize their physical activity and deny that they spend an inordinate amount of time sitting compared to others. This reluctance may be due to concerns about the detrimental effects on overall well-being and feelings of inferiority associated with belonging to the sedentary group [9], consistent with social identity theory [62]. When older adults are asked about their SB, they tend to provide vague estimates but often label their sedentary habits as “excessive” or “substantial” based on their subjective perceptions [58]. To promote effective behavior change, it is important to use positively worded messages rather than accusatory messages to ensure a more receptive response [9].

Several factors may contribute to the negative perception of SB in older adults. First, society places great emphasis on the importance of physical activity for health and well-being, especially in old age [63,64]. This societal emphasis may lead to stigmatization of sedentary habits. Older adults often distance themselves from individuals they perceive as engaging in deviant behaviors in order to maintain control over their own health decisions and demonstrate competence. However, research has shown that emphasizing individual responsibility for health management alone can contribute to stigma and may not effectively promote positive behavior change. To effectively address health behaviors, it is critical to consider the broader influence of social, cultural, and political factors. When these contextual factors are overlooked, it can limit the effectiveness of interventions [65,66]. For example, social norms, cultural beliefs, and systemic barriers can significantly influence individuals’ health decisions and interventions. Therefore, it is important to take a more comprehensive approach that recognizes and considers the interplay between individual responsibility and contextual influences. In this way, interventions can better address the complex dynamics that shape health behaviors and develop strategies that address underlying contextual factors while empowering individuals to make positive changes.

In addition, ageism and negative stereotypes associated with aging influence older adults’ unfavorable views of SB. The societal ideal of an independent and active older adult further reinforces negative perceptions of SB. Research highlighting the negative health effects of prolonged sitting, such as increased risk of chronic disease and functional disability, also may contribute to the negative perception of SB [5,6]. In this context, terror management theory [67] suggests that ageism underlies a psychological mechanism driven by a desire to distance oneself from thoughts of mortality. The desire to distance oneself from sedentary older adults stems from fear of one’s own inevitable decline in health and functional abilities that accompany aging. Consequently, negative attitudes toward SB serve as a means to avoid confronting one’s own mortality. Taken together, these factors shape societal perceptions of SB in older people, leading to widespread misunderstanding and a belief that it should be avoided.

Older adults, however, may respond differently to sedentary activities that involve social interaction or cognitive engagement. Rather than turning away from these behaviors, they are more receptive and open to them. Sitting for social or cognitive activities is perceived as useful and acceptable, whereas purposeless sitting is perceived differently [44]. It is important to note that SB is not only determined by posture, as Voss and colleagues point out [58]. The specific activity performed while sitting also plays a role. Older adults distinguish between sedentary activities, which involve a cognitive activity and are perceived as beneficial, and passive activities such as watching television, the most common form of SB [68,69]. In general, passive activities are perceived as more negative. However, it is interesting to note that even television can be perceived as less passive when it contains educational or news content [58].

In a study by Palmer and colleagues [54], expressed sedentary habits using phrases such as “couch potato” to distinguish the sedentary and passive behaviors portrayed in the media from their own sedentary habits. This finding underscores the importance of distinguishing between passive and mentally active forms of SB in research [10,11,12]. Older adults have a differentiated perception of sitting behavior and categorize it into active and rewarding or passive and deviant. This categorization allows them to retain autonomy over their health decisions and differentiate themselves from perceived sedentary individuals. The findings of Palmer and colleagues’ study [54] illuminate the complexity of older adults’ views of SB and highlight the need to challenge ageist stereotypes and promote a comprehensive understanding of physical activity and SB in the context of aging.

Another interesting observation concerns the way certain older adults perceive sedentary activities. Rather than viewing SB negatively, these individuals consider it as an integral and relatively insignificant aspect of their overall active and engaged lifestyles, thus minimizing its perceived negative effects [9]. In fact, some older adults have positive attitudes about their SB and do not believe it has a detrimental effect on their health [39,43]. For some people, SB serves as a means of relaxation and decompression, especially after prolonged physical activity or work. Older adults who suffer from physical limitations or health problems such as chronic pain or fatigue view SB as an inevitable consequence of prolonged sitting or lying down [49,54]. One possible explanation is that some older adults perceive their bodies as vulnerable and believe their age justifies a sedentary lifestyle [50,54]. They often relate to age and the aging process, distancing themselves from conventional narratives about aging, using them to rationalize their own SB by viewing them as a normal consequence of growing older. This contrast reveals the moralization of sitting, as older adults rank their own SB as acceptable compared to that of others [54].

Eklund and colleagues [41] examined older adults’ perceptions of SB and its importance in relation to health and retirement. Results showed that participants perceived SB differently. On the one hand, SB was perceived as a health risk and something to avoid. On the other hand, it was seen as healthy and beneficial for retirement, which led some participants to change their attitude toward SB. Many participants actively chose to delay the rise of SB by incorporating activities and exercise routines into their post-retirement lifestyles. The study identified three typologies to categorize participants’ descriptions: those who viewed SB as meaningful and desirable, those who experienced SB involuntarily due to illness, and those who associated SB with physical activity and expressed how conscious choices, daily routines, and engagement in physical activity and exercise could delay SB, which they viewed as a natural process of aging.

The results of these studies highlight the complex attitudes and perceptions associated with SB in older adults, which pose a major challenge for intervention development. To effectively address this challenge, it is critical to gain a comprehensive understanding of these nuanced attitudes and tailor interventions accordingly. Future research should focus primarily on exploring older adults’ attitudes and beliefs about SB using qualitative research methods to delve into their experiences and perspectives. In addition, the findings underscore the importance of carefully designing interventions that specifically target SB. Older adults may misinterpret a reduction in SB as an increase in physical activity, which could discourage participation. Therefore, it is important to educate older adults about the clear benefits of reducing SB, regardless of their physical activity level. Using positive wording in intervention messages can increase acceptance and mitigate negative associations [9]. Furthermore, it is important to develop effective behavior change techniques and methods specifically tailored to older adults. In summary, these findings highlight the need for tailored interventions that address misconceptions, provide clear messages, and effectively reduce SB in older adults. When these findings are considered, interventions may be better suited to promote behavior change and improve health outcomes among older adults.

### 4.2. Sedentary Behavior Awareness: Daily Routines and Contexts

Qualitative research plays a critical role in capturing the subjective experiences and perspectives of older people. It offers valuable insights into the factors that influence their daily lives and the contextual elements associated with SB. By exploring these factors in depth, researchers can gain valuable information and develop a deeper understanding of the complexity of SB in this population. This section highlights older adults’ perceptions of their own SB and daily lives.

Although older adults have limited knowledge of the difference between SB and physical inactivity, they can still identify specific sedentary activities that they frequently engage in. In a study by McEwan and colleagues [49], older adults reported spending an average of 5.6 (±1.0) h per day engaged in SB. In addition, older women often lead more active and socially engaged lives compared to men, successfully balancing family, work, and domestic responsibilities, which contributes to higher levels of activity and sociability [50]. To effectively promote and sustain older women’s activity levels, it is critical to address their unique needs through appropriate policies and interventions that leverage their existing roles and responsibilities.

In a study by Palmer and colleagues [54], older adults expressed surprise and disbelief when they discovered the extent of their SB. This finding challenged their self-perception as active individuals and caused some participants to question the accuracy of the monitoring device used in the study, as they felt it did not adequately capture their non-sedentary activities. These reactions can be explained by cognitive dissonance [70], a psychological phenomenon that occurs when individuals experience discomfort or tension due to conflicting beliefs, attitudes, or behaviors. In this case, participants’ awareness of their SB conflicted with their self-perception as active individuals. The presence of cognitive dissonance highlights the challenges people face when confronted with the reality of their SB and the need for behavior change. To address this discrepancy between perceived and actual sedentary time, objective measurement and feedback are critical. Self-monitoring techniques that provide accurate and meaningful data on SB can help older adults overcome cognitive biases and recognize the need for behavior change [71].

Older adults tend to be more sedentary in the afternoon and evening than in the morning. This behavior is influenced by their desire to regulate energy levels and rest after morning activities or save energy for later tasks [39]. Energy level regulation plays an important role in prolonged sitting during these periods. In addition, declining physical performance and ingrained social habits also contribute to this behavior pattern. To counteract SB in older adults, it is important to develop strategies that encourage non-sedentary activities during the day that can help break up prolonged periods of passive sitting. By incorporating more active pursuits into their daily routines, older adults can interrupt sedentary time, increase energy expenditure, and improve their overall physical well-being [54].

In recent years, the field of time-use epidemiology has evolved with methodological advances. This approach considers physical activity, SB, and sleep as interrelated components of the 24 h day rather than independent risk factors [72,73]. By viewing these components as mutually exclusive and exhaustive, researchers can investigate how the reallocation of time between physical activity, SB, and sleep impacts health and determine the optimal balance for overall well-being [72]. This is an opportunity for future research to use the principles of time-use epidemiology, which may provide valuable insights into the interplay between physical activity, SB, and sleep, and their effects on health.

In general, older adults engage in a variety of sedentary activities, both individually and in their communities. While watching television is the most commonly reported sedentary activity among older adults, they also engage in a number of other sedentary leisure activities, including solitary pursuits such as reading, doing crossword puzzles, knitting, and crocheting [39,49,52,57]. Other activities include listening to music, working on the computer, and playing musical instruments [49,53,57]. In addition, they participate in group activities such as bingo or card games [39,49,52]. These findings highlight the complex nature of SB in the elderly, which involves a mix of individual preferences and social dynamics. They offer valuable insights into the motivations and preferences of older people with respect to SB. Participation in solitary activities provides people with personal pleasure and mental stimulation while remaining sedentary. Conversely, participation in group activities promotes social interaction and a sense of community among older people, even if these gatherings tend to be sedentary [39].

In their study, Nuwere and colleagues [52] studied the time allocation of older adults in different domains of SB. Leisure activities were found to occupy the most time, while work-related activities occupied the least. The most common sedentary leisure activities included TV, online activities, and socializing with friends, with an average duration of 2.6 h, 2.1 h, and 2.1 h per day, respectively. Older adults also reported spending approximately 1.7 h per day sitting on public transportation or in a car to commute. Sedentary activities in the home environment included taking meals at home for about 1.8 h per day and completing administrative tasks for about 1.4 h per day. At work, computer use was the predominant sedentary activity, averaging 0.5 h per day.

Palmer and colleagues [53] examined sedentary and non-sedentary behaviors in a variety of contexts, including leisure, household, transportation, and work. Most sedentary leisure activities, such as watching television, reading, doing crossword puzzles, and using technological devices, were predominantly done at home. However, some sedentary activities also occurred in cafes, restaurants, bars, theaters, and movie theaters. Non-sedentary activities such as shopping and museum visits, on the other hand, were more often performed outside enclosed spaces. The study found gender and socioeconomic differences in SB, with women showing higher participation in sedentary activities outside the home, such as café and theater visits, and in sedentary hobbies such as knitting, sewing, and playing bingo. Conversely, men tended to spend more time in non-sedentary activities outside the home, particularly sports. In the household domain, all participants reported engaging in non-sedentary activities. Several emphasized the importance of incorporating household chores into daily routines to increase physical activity and minimize prolonged sitting. Both higher- and lower-activity participants acknowledged the importance of getting around actively, such as walking or using public transportation, although they also admitted to using less active means such as driving their own cars. In addition, participants with less SB reported greater engagement in sedentary and non-sedentary occupational activities compared to participants with higher SB. These findings highlight the need to consider multiple factors that influence these behaviors and may help identify specific sedentary activities that can be used for interventions to reduce SB.

According to Chastin and colleagues [39] older adults are more likely to get up in the morning to do chores or go for a walk. Motivation for upright behavior in older adults included six key factors: alleviating physical discomfort, combating boredom and depression, asserting independence and self-esteem, participating in social and recreational activities, personal characteristics, and environmental influences. Study participants acknowledged the importance of maintaining an upright posture and actively participating in activities, both for self-care and for caring for loved ones. They also recognized the need for external incentives from the community, social environment, and family to promote upright behavior, while emphasizing the importance of a safe environment.

Given the unique daily routines and contextual factors relevant to older adults, it is possible to adapt interventions to the specific challenges of this population. These interventions can include a variety of strategies, such as promoting active alternatives to sedentary activities during specific time periods, encouraging social engagement that includes physical activity, and modifying the home environment to allow for more physical activity while reducing opportunities for prolonged sitting. The valuable findings from the qualitative research serve as the basis for developing interventions that are contextual and tailored to the needs of older adults, with the ultimate goal of reducing SB in this population.

### 4.3. The Dual Nature of Sedentary Behavior: Perceived Benefits and Drawbacks

A review of the qualitative literature on SB among older adults uncovered another theme related to perceptions of the advantages and disadvantages of being sedentary. Older adults were found to have unique perspectives and motivations that shape their participation in SB, and their self-perceived level of activity or inactivity has a profound impact on their interpretations. In addition, their understanding of SB was influenced by the cultural and social context in which they live, as well as their awareness of the health implications associated with it [54]. Therefore, it is critical to explore their perspectives in order to develop effective and sustainable approaches that promote active aging and improve overall health outcomes.

Overall, older adults perceive the negative effects of SB primarily on their health, including poor mental health, social isolation, increased risk of disease, loss of muscle mass and strength, poor circulation, and decreased independence and mobility [9,39,49,52,58]. Perceptions that excessive sitting is unhealthy are primarily associated with short-term consequences such as increased pain and stiffness, difficulty getting up from a seated position, and risk of falling asleep while sitting [39,49,52,56]. Older adults also express concern about long-term consequences, including weakened bones and muscles [52], impaired heart health, and weight gain [56]. These physical limitations are attributed to the natural aging process and are recognized as factors that prevent them from engaging in more active activities [49]. In addition, older women are often encouraged or pressured to sit more than they would like, while their family, friends, and caregivers often discourage them from physical activity [39].

It is, therefore, imperative to raise awareness of the health risks associated with SB and educate people about the methods and benefits of reducing this behavior. However, Leask and colleagues [46] emphasize that people who are unaware of or do not fully understand the consequences of their SB are unlikely to be motivated to reduce it. It is important to recognize that not all SB should be demonized, as some sedentary activities can be useful and enjoyable for individuals. A nuanced approach that recognizes the value of SB while promoting a balanced and active lifestyle will likely be more effective in achieving behavior change.

Interestingly, McGowan and colleagues [9] observed that older adults are aware of the reciprocal relationship between prolonged SB and its negative effects on well-being. Participants recognized that factors such as depression and poor health may contribute to their tendency to sit and spend time indoors. However, they also acknowledged that continued inactivity further exacerbated their depression and health problems, creating a self-reinforcing cycle. While participants showed understanding of the potential benefits of reducing SB, there was some ambiguity between reducing SB and increasing physical activity. When it came to reducing SB, they cited several benefits, including improved social interaction, mobility, strength, lower blood pressure, improved circulation, distraction from pain, improved sleep quality, increased mental well-being, and improved cognitive performance. Participants who were already very physically active, however, saw fewer benefits of reducing SB alone and believed that simply moving around the house would not provide significant benefits. Nevertheless, some of these individuals were willing to reduce their SB if they were convinced of the additional benefits. These findings highlight the need to clarify the specific benefits of reducing SB and to address any misconceptions among individuals who are already physically active.

Older adults, while acknowledging the potential health consequences of prolonged SB, also view it as a necessary and useful coping strategy, citing positive aspects such as pleasure, mental stimulation, and social contact [9,39,52,58]. By incorporating SB into their daily routines, they effectively manage symptoms of chronic disease, restore energy levels, relax, and promote mental well-being [9,39,52]. Older adults also recognize the physical benefits of participating in activities at senior centers, such as walking across parking lots or engaging in physically demanding aspects of daily life [49].

Engaging in sedentary activities also provides older adults with opportunities to meet new people and make friends. Social engagement is seen as a way to prepare for the future, especially for those who are aging and no longer have children at home. The social aspect of sedentary activity plays a critical role in providing companionship and support during life transitions [49]. These findings underscore the importance of considering the benefits and meaningfulness of SB when developing interventions to reduce SB among older adults.

In addition, sedentary activities give structure to the day, especially after retirement [52]. Older adults emphasize that any form of activity, regardless of its type, gives them a sense of purpose and motivation to start their day and fill their time. These activities can provide cognitive and physical benefits and give meaning to their daily lives [49]. Older adults tend to prefer busyness over idleness and associate negative traits such as laziness and passivity with prolonged sitting. They actively distance themselves from the idea of being lazy and emphasize avoiding passive sitting. These findings suggest that older adults rationalize and justify their sitting behavior based on cultural norms and personal values. They emphasize the importance of engaging in meaningful activity while sitting and discourage prolonged passive sitting [54].

One approach to address this issue is to develop guidelines that consider the lived experiences and expertise of older people and move beyond a simple understanding of sitting based on epidemiological evidence. These guidelines should move away from viewing SB as the opposite of physical activity. Instead, it may be more beneficial to understand how sitting is part of a broader pattern of movement throughout the day [74] and what factors shape these patterns, such as energy levels, enjoyment of sitting, social structures, and cultural expectations [53]. Counseling should therefore emphasize the positive aspects of sitting, such as recreation, pleasure, and cognitive benefits, while encouraging exercise and finding an individualized balance between sedentary and non-sedentary activities that can be integrated into daily routines [75].

Overall, these findings underscore the complex relationship between SB and its consequences, highlighting both benefits and drawbacks. This emphasizes the importance of interventions that address the negative impacts of SB while meeting people’s needs for recreation and enjoyment. It is important to consider this point of view when developing policies, as simply banning sedentary activities such as playing cards or watching educational programs may not be well received. Encouraging frequent breaks during inactive time may prove to be a more successful strategy. Interventions can be better adapted to the needs and preferences of older people by finding a balance between minimizing SB and short periods of recreation [58].

### 4.4. Delving into Sedentary Behavior: Determinants and Discouraging Factors

Analysis of the factors that either discourage or promote SB in older adults is a topic of great importance in the field of aging and health research. Understanding these factors can provide invaluable insight into the complicated nature of SB in older adults and ultimately aid in the development of interventions to promote more active lifestyles in this population. Given the growing recognition that prolonged sitting has detrimental health effects and the increasing prevalence of a sedentary lifestyle in older adults, it is imperative to further explore the underlying influences that contribute to SB. By uncovering the complex interplay of these factors, researchers and clinicians can develop effective strategies to address SB and facilitate meaningful behavior change in older adults. Consequently, this comprehensive understanding holds great potential for promoting successful aging and reducing the burden of chronic disease often associated with SB.

#### 4.4.1. Determinants

SB in older adults is influenced by a variety of factors that include personal, interpersonal, and environmental dimensions [39]. Recognition of these distinct categories is consistent with socio-ecological models [76] that emphasize the complex interplay between behavior and its surrounding influences [17,77]. Promoting active lifestyles and reducing SB require a comprehensive approach that considers the multifaceted nature of SB.

In a personal context, physical health problems contribute to the prevalence of SB in older adults [39,43,50,52]. For example, chronic diseases such as arthritis often trigger SB, as symptoms such as pain and stiffness discourage physical activity and encourage SB [39]. Fatigue and low energy levels may also discourage physical activity and contribute to SB. Another factor is a lack of awareness of the negative health effects of prolonged sitting and the importance of regular physical activity in daily life [43]. This lack of awareness may be further influenced by mobility problems, cognitive decline, and depression, which affect decision making regarding physical activity [50,52].

Other factors influencing SB include a lack of motivation, a tendency toward laziness, difficulty interrupting sedentary activities with short breaks, and fear of falling or injury [44,49,51]. Negative attitudes about aging also play a role in SB [44,49,51,55]. Financial costs, enjoyment of sitting, mental stimulation from sedentary activities, ingrained habits, and limited knowledge of available programs are other personal factors that may contribute to SB [43,55,59]. These personal factors significantly influence the decision-making process and may contribute to the adoption of SB.

In the interpersonal domain, negative age-related stereotypes associated with decline, devaluation, and passivity have been identified as factors promoting SB in older adults [39,49,51,55]. Lack of companionship and overworked caregivers are also SB determinants. In addition, loss of social contact due to health problems or relocation of friends may lead to SB and social isolation [50,53]. While social norms and everyday tasks such as housework and caregiving encourage standing activities that promote self-efficacy and independence, concerns about burdening family members or caregivers may limit older adults’ engagement in these activities. Furthermore, negative attitudes about aging and societal perceptions of seniors, including the label “senior”, may discourage older adults from being more active [49,52,55,59].

To promote positive outcomes in the self-assessed health of older adults, it is important to implement person-centered goal setting that empowers them to make decisions about their preferred activities. By targeting the specific areas of their daily routines where they are most likely to reduce SB, individualized approaches can be effective. It is important to recognize and respect older people’s autonomy in choosing their activities so that interventions can be tailored to their individual needs and preferences. This person-centered approach has the potential to lead to better outcomes in promoting their overall well-being [78].

Reducing SB can also be challenging due to time constraints resulting from personal or interpersonal issues. Many people struggle to find time for exercise amid their busy schedules, caregiving responsibilities, and other commitments. Similarly, social pressures and prevailing societal norms often prioritize sedentary activities over active ones, making it difficult for people to prioritize physical activity in their daily lives. Lack of support may further hinder efforts to adopt a more active lifestyle. Without the necessary encouragement, guidance, or a supportive environment, individuals may find it difficult to persevere with behavior change and overcome barriers. For older adults, caregiver interactions and social attitudes have a significant impact on SB. Supportive caregivers and a strong social network can significantly improve older adults’ motivation and ability to reduce their SB [43]. Cultural influences and desire for acceptance also impact older adults’ choices and often lead them to engage in more traditionally defined SB [50,55].

Webber and colleagues [59] conducted a qualitative study to examine the factors that influence SB in individuals with osteoarthritis or after total knee arthroplasty. The study found that participants’ SBs were more influenced by social, cultural, and environmental factors than by personal characteristics. Participants cited workplace culture and societal expectations as barriers to reducing SB. In addition, the study revealed a gap in knowledge regarding the health risks associated with SB and the importance of incorporating light physical activity into overall physical activity levels. These findings underscore the need for comprehensive programs that address social, environmental, and knowledge-related factors to promote more physical activity and less SB in individuals with osteoarthritis or after knee surgery. Raising awareness of these factors is an important first step in implementing effective behavior change interventions that can lead to better health outcomes, less SB, and improved overall well-being.

Environmental factors also play a critical role in the development of SB in older adults. Several studies have identified key factors contributing to SB, including a lack of community-based activities, facilities, and services that promote standing and physical activity [37,39,50,51,52,53,55]. Older adults are often dissatisfied with the limited opportunities to be active and the frequency of activities that require them to remain sedentary [39]. Financial costs, inadequate subsidies for senior programs, and limited access to programs are other determinants of SB [41,49,55]. Limited availability of physical activity programs is a major challenge and discourages many older adults from participating in classes and activities. Older adults emphasize the need for better information dissemination about available programs in their communities and point to the importance of primary care physicians and the media in promoting active lifestyles [50,55]. By addressing these environmental factors and improving access to programs and information, we can effectively reduce SB in older adults.

Other environmental factors such as weather conditions and urban design also influence the amount of time spent sitting. Inclement weather, limited availability of public seating, and physical barriers such as residential stairs and cracked sidewalks have been identified as contributing factors to SB [51,52,54,56,58]. These factors present challenges and safety risks that discourage active participation, such as waiting for transportation in inclement weather [39,49,52,55,58]. Age-related limitations on driving and limited accessibility to public transportation further discourage older adults from engaging in physical activity outside of their homes [50,55,58]. These environmental determinants are consistent with those identified in the existing literature on physical activity [79].

Furthermore, the home environment and organizational settings in residential facilities often discourage individuals from engaging in activities of daily living, creating additional barriers to movement [38,44,58]. The home environment plays a role in limiting daily activities of low to moderate intensity, such as washing, cooking, and moving from room to room, which contributes to an increasingly sedentary lifestyle among older adults. Certain aspects of the home, such as stairs, available space, and facility design, are identified as particularly limiting factors associated with SB [38].

Recent research shows that older adults living in assisted living facilities often lead sedentary lifestyles and stand and walk less compared to their independently living peers [80]. Interestingly, social participation among older adults has been associated with reductions in passive SB and increased physical activity [81]. These findings underscore the importance of promoting and supporting community engagement in assisted living environments, as it can improve both cognitive and physical functioning in older adults. By promoting social participation, we can mitigate the negative effects of SB while improving the overall well-being of this population.

In a study conducted by Kotlarczyk and colleagues [44], older adults in assisted living facilities indicated that the living environment contributed significantly to their SB. Moving into a residential facility led participants to perceive their lives as more sedentary and attributed this to the presence of amenities and services that limited their engagement in home activities. Although facilities offered a variety of activities, in many cases participants were required to sit for extended periods of time. The physical environment also influenced participants’ inclination to engage in SB. They expressed concerns about going outside and suggested measuring distances in the hallways. In assisted living facilities, caregivers were able to provide physical assistance to residents with limited mobility, while in independent living facilities, limited staff availability made it difficult to reduce the amount of time spent sitting. This study provides insights into how older people perceive and experience SB in shared living arrangements and highlights the influence of the living environment. The findings emphasize the importance of addressing physical limitations, promoting social engagement during sedentary time, and addressing environmental factors to effectively reduce SB.

The analysis of the factors that determine SB in older adults is consistent with the Capability, Opportunity, Motivation, and Behavior (COM-B) model [82]. This model emphasizes the role of individual capability (both physical and psychological), opportunities (both social and physical), and motivations (both automatic and reflective) in shaping behavior. Regarding capabilities, age- and health-related decline in physical ability has been identified as a major cause of SB among older adults. This decline limits their ability to stand for extended periods of time or engage in moderate to vigorous physical activity. In addition, there is a lack of awareness of the negative health effects of prolonged sitting. The opportunity component of the COM-B model emphasizes the influence of the physical, cultural, and social environment on behavior. Environmental factors, including weather conditions, were also found to play a role in influencing SB. These findings demonstrate the importance of considering external factors in addition to individual motivation and capability when attempting to reduce SB. By considering environmental factors, such as providing indoor alternatives for exercise during inclement weather or creating comfortable outdoor spaces, people can be encouraged to be more active. Finally, automatic motivation, which is determined by habits and routines, is considered one of the main causes of SB in older adults. Interventions should therefore acknowledge the habitual nature of SB and incorporate strategies to interrupt these habits, such as modifying environmental stimuli and raising awareness through feedback and self-observation techniques [71].

Overall, these findings underscore the importance of addressing individual, interpersonal, and environmental factors to reduce SB. Strategies such as raising awareness of the harmful effects of prolonged sitting, creating opportunities for physical activity despite lack of time, challenging social norms, addressing energy-related issues, and fostering a supportive environment are critical to promoting behavior change in older adults [43,55]. The qualitative research conducted highlights the complex interplay of individual, social, and environmental factors in shaping SB in this population. It also stresses the need for public policies that promote access to and availability of senior centers to support healthy aging among older adults.

#### 4.4.2. Discouraging Factors

Reducing SB in older adults is a multifaceted process influenced by several factors. One critical aspect is the varying opinions and attitudes of older adults about the benefits of reducing SB. While some individuals express willingness and interest in reducing SB, others may believe that their age entitles them to maintain a sedentary lifestyle. Skepticism about the benefits of standing more and concerns about disrupting daily routines are also common. However, amidst these differing opinions and concerns, it appears that many older adults have a clear interest in changing their sedentary habits and incorporating more standing activities into their daily routines while they still have the ability to prioritize rest and flexibility [39].

Motivators for reducing SB in older adults include a desire for better health, increased awareness of SB, ease of integrating changes into daily life, and a desire to maintain independence and functionality [40,43,55]. Older adults place a high priority on remaining mentally and physically active to maintain overall well-being. Sedentary activities that provide social interaction, cognitive stimulation, or relaxation are highly valued, whereas passive television viewing and purposeless sedentary activities are considered of little value [53]. Participation in meaningful activities and a sense of purpose are considered essential [50,53]. Enjoyment of non-sedentary activities, social support, companionship, intellectual stimulation, curiosity, and motivation are also cited as reasons for participating in such activities [37,40,43,50,53,55].

The social and physical environment has a significant influence on SB in older adults. Research shows that affluent neighborhoods that offer more opportunities for community engagement tend to have lower rates of SB than disadvantaged neighborhoods with limited opportunities. In addition, social support and a sense of connection to the community play a critical role in encouraging older adults to participate in non-sedentary activities [50]. For example, participation in classes provides social interaction, builds friendships, and offers mental stimulation, all of which help reduce SB [40,55,58]. Maintaining autonomy and self-determination is also critical, and having a caregiver or support network helps older adults overcome dependence and reduces SB [51]. Meaningful activities such as gardening or volunteering have been shown to discourage SB in older adults [55]. Furthermore, improving the physical environment, such as infrastructure and home size, is critical to promoting an active lifestyle in older adults [50].

Strategies commonly cited by older people to reduce SB include creating opportunities for social engagement, ensuring safe environments and transportation, tailoring activities to individual preferences, and fostering a sense of caring for others. Alternatives to sitting, such as gardening, dancing, or shopping, are considered more effective than focusing solely on reducing sitting. Support from family, friends, and peers is considered beneficial. In addition, short, simple activities that can be easily integrated into daily routines are preferred [39]. Gradual changes, using existing resources, and adapting activities to individual abilities are also emphasized. Modifying the physical environment by providing well-maintained pathways, rest areas, and stimulating green spaces can also help reduce SB [50].

Promoting active aging and reducing SB among older adults can be supported by a range of interventions that focus on increasing the availability, accessibility, and awareness of key resources. Effective strategies include removing financial barriers, improving transportation options, creating an age-friendly physical environment, and fostering social relationships [50]. A comprehensive social-ecological approach [76] is critical to addressing SB in older adults because it recognizes the interplay of multiple factors that influence behavior. Elements such as the physical environment, social engagement, community meals, organized activities, access to transportation, and community involvement all play important roles in reducing SB [58]. Proximity and quality of recreational opportunities also have a significant impact on active lifestyles, underscoring the importance of supporting and funding senior centers and other community facilities that promote healthy aging. In addition, interventions should consider personal beliefs, societal factors, and life roles as they influence older adults’ attitudes toward physical activity [41,52]. By understanding and addressing these factors, tailored interventions can be developed to promote active lifestyles and reduce SB in older adults.

In summary, when examining the factors that influence SB in older adults, it is important to consider a number of aspects. These include exploring multiple perspectives, understanding individual motivations, recognizing the importance of meaningful activity, examining the influence of the social and physical environment, acknowledging support systems, and considering personal beliefs. Interventions that are tailored to individuals’ preferences, provide enjoyable alternatives to sitting, and consider the barriers and facilitators within the social-ecological framework [76] have been shown to reduce sedentary time and promote active lifestyles in older adults [58].

### 4.5. Breaking the Sedentary Spell: Interventions

While most research on SB has focused primarily on quantitative studies, there are a limited number of qualitative studies that have addressed the intricacies and complexities of interventions targeting SB in older adults. These qualitative studies address three critical aspects of SB interventions: the motivators and barriers that influence older adults’ participation in SB reduction programs, the outcomes of these interventions, and older adults’ suggestions for improving interventions. By examining the subjective experiences, perceptions, and contextual factors associated with SB interventions, these studies provide insights that complement quantitative data. In this section, we summarize key findings from qualitative studies that specifically examine SB interventions. We highlight the motivators and barriers that influence participation in these programs, the impact of these interventions on older adults’ levels of SB and overall well-being, and their suggestions for intervention development.

#### 4.5.1. Barriers and Facilitators to Participating in Sedentary Behavior Interventions

Older adults’ participation in SB reduction programs is influenced by a number of barriers and factors that greatly affect their interest in such interventions. Therefore, it is critical to understand these factors in order to develop effective strategies to promote active aging and successfully reduce SB in this population. Some studies have examined these factors to gain insights and develop evidence-based approaches to promote sustained participation and achieve positive outcomes in SB reduction programs for older adults.

Participation in SB reduction programs may be hindered by several obstacles. One major barrier is fear of physical limitations and age-related safety concerns. Older adults may be concerned about their health or have physical limitations that prevent them from participating in SB reduction programs. They fear that participation in such programs could lead to injury or exacerbate their existing health problems. This fear stems from the idea that their bodies may be more vulnerable to strain or injury as they age [50,54]. Studies have shown the impact of these fears on older adults’ willingness to participate in SB reduction programs [42,47,48,54,60]. These studies highlight the importance of addressing these concerns and ensuring the safety and appropriateness of program activities for older adults. By addressing these fears and providing appropriate guidance and support, SB reduction programs can help older adults overcome this barrier and enable them to lead more active lifestyles.

Lack of motivation, a strong preference for sedentary activities, and deeply ingrained sedentary habits are also significant barriers for older adults to reduce their SB. Some studies have shown that it is difficult to motivate older adults to change their SB because they enjoy sedentary activities such as reading, watching television, or hobbies that require little physical exertion [42,43,47,48,60]. Therefore, it is important to find alternative, non-sedentary activities that are enjoyable and personally meaningful to older people. By providing such activities, we can encourage them to replace sedentary activities with more active ones without feeling deprived or unfulfilled.

Additionally, lack of time and family obligations hinder older adults’ participation in SB interventions. They often have busy schedules that revolve around caring for loved ones, maintaining social relationships, or meeting other family obligations. These time constraints and family obligations limit their time and energy for actively participating in SB reduction programs or incorporating physical activity into their daily routines [42,43,47,48,60]. Recognizing and addressing these time and family constraints is critical in SB interventions that align with the realistic and feasible demands of older adults’ busy lives. Providing flexible options, integrating physical activity into daily tasks, and offering support systems that accommodate family responsibilities can help overcome these barriers and encourage older adults to lead more active lifestyles.

Limited social support and a lack of a support network may further hamper older adults’ participation in SB reduction programs. One of the biggest challenges is that older adults may not have friends or family members who share similar goals and interests in reducing SB. The lack of a support network makes it difficult for them to find guidance and encouragement on their SB reduction journey, resulting in lower motivation and poorer adherence to the program. In addition, the lack of a support network can lead to a lack of accountability, as there is no one to check progress, celebrate successes, or provide gentle reminders when motivation wanes [42,43,47,48,60]. To overcome this barrier, it is important to create opportunities for group activities such as group classes or community events, as these can serve as a platform for reducing sedentary time and fostering social relationships. In addition, fostering intergenerational interactions can provide older adults with a sense of purpose and fulfillment while offering valuable opportunities for socialization and support. These strategies can help address the limited social support older adults experience and improve their overall well-being on the path to reducing SB.

Societal expectations and norms may also act as barriers to reducing SB in older adults. The prevailing perception that SB is acceptable and appropriate for older people may contribute to a lack of motivation or social pressure to maintain a sedentary lifestyle [42]. Furthermore, limited knowledge about the negative health effects of prolonged sitting and the benefits of an active lifestyle may affect older adults’ awareness and understanding of the importance of reducing SB [47]. As a result, older adults may be reluctant to change their behavior and underestimate the importance of reducing SB.

Environmental barriers may pose an additional challenge for older adults to participate in SB reduction programs. Limited access to recreational facilities may prevent them from engaging in physical activity and finding appropriate alternatives to sitting [43]. In addition, inadequate infrastructure, such as poorly maintained sidewalks or lack of wheelchair accessibility, may limit their mobility and discourage active behaviors [48]. Extreme weather conditions, including excessive heat or cold, can further limit opportunities for outdoor activities and make it difficult for older adults to reduce SB [74]. Furthermore, neighborhood characteristics such as high crime rates or inadequate green spaces may raise safety concerns and discourage older adults from engaging in SB reduction interventions [43].

Another barrier for older adults is limited access to professional educators who can provide guidance and support. The availability of qualified professionals who can guide and support older adults in adopting healthier behaviors may be limited, especially in certain communities or health care settings. This lack of guidance and support may prevent older adults from overcoming barriers and implementing effective strategies to reduce their SB [48]. To address these challenges, it is important to increase access to recreational facilities, improve infrastructure to promote mobility, consider weather conditions when designing programs, promote safe and supportive neighborhood environments, challenge social norms related to SB in older adults, raise awareness of the risks of prolonged sitting, and ensure the availability of qualified professionals to provide guidance and support.

In addition to the barriers already mentioned, several factors have been identified that promote participation in SB reduction programs. These factors have a significant impact on people’s motivation and willingness to participate in such programs. One critical factor is awareness of the health benefits associated with reducing SB. When people are informed about the negative effects of excessive sitting and the potential health risks associated with it, they are more likely to act. Knowledge of the association between prolonged sitting and various health problems is a strong motivator for people to actively engage in SB reduction programs [42,47,48,54].

Social support is widely recognized as a critical factor supporting people’s efforts to reduce their SB. Studies have shown that supportive relationships with family members, friends, or peers who participate in active behaviors have a major impact on motivation and adherence to SB reduction programs [42,43,47,48]. These relationships provide valuable elements such as encouragement, accountability, and a sense of belonging, and make the process of behavior change more enjoyable and sustainable. Therefore, it is important to incorporate social support strategies into interventions to promote active lifestyles in older adults [9,39,40,55,58]. Group-based activities, buddy systems, and other approaches that promote the development of support networks can increase the effectiveness of such interventions.

The accessibility and convenience of programs are also critical factors. Programs that are easily accessible in terms of location, time, and affordability are more likely to attract and retain participants. The convenience of programs allows individuals to seamlessly incorporate exercise into their daily routines without major interruptions. Interventions that promote active alternatives to sedentary activities, such as integrating opportunities for physical activity into the home environment or incorporating standing or walking breaks into sedentary activities, can effectively support behavior change [39]. By emphasizing accessibility, convenience, and active alternatives, programs can better support individuals in adopting and maintaining a physically active lifestyle.

In addition, engagement and participation in SB reduction programs can be increased by providing activities that are enjoyable and have personal meaning. When interventions are tailored to individual preferences and provide opportunities for social interaction and cognitive engagement, this can increase intrinsic motivation and boost participation rates. These findings are consistent with self-determination theory [83,84], which emphasizes the importance of autonomy, competence, and connectedness for behavior change [51,78]. By identifying and addressing individual needs and preferences, programs can be tailored to different fitness levels, interests, and goals, maximizing their effectiveness.

Other factors that contribute to the success of SB reduction programs include a sense of personal responsibility, regular activity, improved self-esteem, and support from qualified professionals without financial burden [42,43,47,48]. When people feel personally responsible for their health, engage in regular physical activity, experience better self-esteem by reducing sedentary activities, and receive guidance and support from professionals without financial barriers, SB programs can effectively attract and retain participants. This leads to increased engagement and long-term adoption of an active lifestyle.

In summary, identifying the factors and barriers to participation in SB reduction is critical to developing interventions that effectively address these factors. Creating a supportive social environment, offering tailored and accessible programs, and addressing time constraints and perceived limitations can increase participation rates and promote long-term behavior change. By identifying and addressing these facilitators and barriers, program designers and policymakers can develop strategies that encourage and empower individuals to adopt and maintain a physically active lifestyle. To develop effective interventions against SB, it is important to consider the specific context in which this behavior occurs [85]. In addition, behavior change interventions should consider multiple levels of the social-ecological model [76]. By considering the individual, interpersonal, community, and societal factors that influence SB, interventions can be tailored to effectively promote behavior change and create an environment conducive to SB reduction.

#### 4.5.2. Impacts of Sedentary Behavior Interventions

Qualitative research examining the experiences of older adults in SB reduction programs has provided valuable insights into the factors that contribute to their engagement and success in these interventions. A key finding from these programs is that participants have developed a greater awareness of the negative consequences of SB, which has led to greater motivation to make positive lifestyle changes and find simple ways to reduce sitting time. Realizing the importance of being active and the impact SB has on their health was a driving force for their participation [42,43,47,48]. Reminders played a critical role in keeping older adults aware of their SB and encouraging them to take regular breaks [47].

In a study led by Greenwood-Hickman and colleagues [43], overweight and obese older adults participated in a program that resulted in significant changes in their daily routines and exercise habits. Participants reported spending more time standing, especially during sedentary activities such as watching television. They actively built in breaks from sitting, which led them to engage in various activities and chores throughout the day. In addition, some participants incorporated additional physical activities into their daily routines, such as taking walks or participating in exercise classes. One notable outcome of the program was the participants’ heightened awareness of sedentary activities. They became more aware of the amount of time they spent sitting and standing, allowing them to make informed decisions about their SB habits. These results underscore the positive impact of the program in promoting a more active lifestyle among older adults.

Participation in these SB reduction programs had a positive effect on the physical and mental health of the older adults. They further perceived the health benefits of the program, supported by their belief in its effectiveness. Reported improvements included higher energy levels, mobility, flexibility, strength, pain management, reduced stiffness, balance, sleep quality, and overall physical fitness. In addition, participants highlighted positive effects on mental well-being, such as lower stress levels, improved mood, increased confidence, better memory and concentration, improved cognitive function, fewer depressive thoughts, and easier recovery from negative moods [42,43,47,48]. Positive attitudes and perceived health benefits indicate the value of these interventions for improving physical function, self-confidence, and autonomy in older adults [48]. These findings underscore the wide-ranging benefits of interventions aimed at reducing SB.

In addition to the physical benefits, social support and a sense of engagement within these programs emerged as critical factors for older adults. The group settings created a supportive and inclusive environment in which participants felt a sense of belonging. They valued the opportunity for social interaction and formed meaningful friendships with both their peers and program leaders [43]. This social support network played an important role in maintaining their motivation and commitment to the program as participants received encouragement and inspiration from their peers and leaders. Participation in these programs not only promoted social contact, but also counteracted social isolation as older adults participated in group activities, interacted with peers, and pursued common goals [43,48,60]. Furthermore, participants became more aware of the social norms that promote excessive sitting, which led some to change their socialization patterns and adjust the amount of time they spend with others, ultimately reducing SB [43].

Tailored and enjoyable activities were highly valued by participants as they contributed to sustained engagement and motivation. Participants appreciated interventions that offered a variety of activities tailored to their specific needs and preferences, allowing them to select activities that matched their interests and abilities. This personalized approach increased their satisfaction with the programs and the overall experience. These findings underscore the importance of a personalized approach to intervention acceptance and participation. Positive attitudes toward the interventions and perceived health benefits suggest that they can be effective in improving physical function, self-confidence, and autonomy in this population [48]. These results also demonstrate the importance of a supportive social environment and personal accountability in facilitating behavior change related to SB [47].

Taken together, these qualitative findings underscore the importance of developing SB reduction programs that raise awareness, provide social support, and offer tailored and engaging activities. Taking these findings into account, program designers and health professionals can develop interventions that better meet the needs of older adults and promote sustained engagement to reduce SB.

#### 4.5.3. Interests and Preferences for a Sedentary Behavior Intervention

Recent studies have begun to examine older adults’ perspectives on interventions to reduce SB. While previous research has focused primarily on quantitative measures, qualitative studies offer valuable insights into the suggestions and preferences expressed by older adults themselves. This section summarizes the findings of qualitative studies that have examined older adults’ interests and preferences regarding interventions to reduce SB.

The results of the studies highlight important recommendations from older adults on strategies to reduce SB. Tailored interventions that address individual abilities emerge as important [42,45,46,47,48,56]. Older adults emphasize the need for personalized approaches that consider their unique circumstances and motivations. They recognize the significant differences between different situations, such as age, and emphasize the need for multiple solutions to meet different preferences [45]. A tailored approach not only boosts their confidence and motivation to engage in physical activity, but also reduces the risk of injury [53]. In addition, older adults appreciate programs that are flexible in time and location, as this makes it easier for them to incorporate physical activity into their daily routines [46,47,48,56]. They also emphasize the importance of using existing resources in their daily lives to reduce SB. These resources include physical and mental skills, interpersonal support, knowledge, and opportunities for occasional activities [45].

Leask and colleagues [46] found that an intervention designed as an extension of an everyday object proved highly effective. The co-creators (older adults) of the intervention adapted it to their daily lives, providing congruence and seamless integration. The intervention consisted of several key elements. First, it included an education component to address older adults’ lack of awareness of the health consequences of SB. Second, the intervention included simple solutions that could be easily incorporated into daily life with the goal of interrupting sedentary activities. Third, self-monitoring tools were provided to raise awareness of SB and enable users to make changes that were compatible with their lifestyle. The intervention emphasized autonomy and allowed users to determine the frequency of self-observation through a diary format. Important information to note included the longest time of sedentary activity during the day, the purpose of the prolonged sedentary activity, and the reasons for interrupting these times to provide users with contextual information. Fourth, participants were asked to set goals, with a focus on interrupting the two longest sedentary activities to ensure the feasibility and sustainability of change. Finally, users had the opportunity to document the benefits they experienced from interrupting SB to encourage long-term retention of the intervention.

Overall, it is paramount to develop tailored interventions that fit seamlessly into the lives of older adults and provide them with the education, resources, self-monitoring tools, goal setting, and assessment they need to effectively reduce their SB [46]. By applying the COM-B model [82] in this context, one can gain insight into the various factors that contribute to behavior and explore the assets that individuals possess in terms of their abilities, opportunities, and motivations. For example, in the context of SB, individuals have the ability to interrupt sitting if they want to reduce their stiffness or if they do not value a particular sedentary activity. The ability to interrupt SB can take several forms, including social options such as getting up to answer the door or physical options such as taking medication. These findings underscore the importance of considering multiple options appropriate to the context of SB. Providing a range of options that can be used at different times of the day and for different reasons can help people reduce SB.

Educational components also play a critical role in interventions. Research studies have shown the importance of providing information about the health benefits associated with reducing SB and offering practical tips and strategies to incorporate physical activity into daily life [45,46]. Older adults express that they need this knowledge because it enables them to make informed decisions and take control of their own health. When they understand the potential positive effects of reducing SB and have access to practical guidance, older adults are more likely to be motivated and able to change their sedentary habits. Educational components provide the necessary foundation for older adults to actively engage in SB reduction programs and increase their self-efficacy and confidence in healthier behaviors. It is important to educate people about the health risks and benefits of reducing SB, but this should occur in a way that considers their understanding and motivation. Simply demonizing all forms of SB is unlikely to be effective, as some sedentary activities may have cognitive benefits. It is important to distinguish between active, purposeful, and passive sedentary activities, as individuals can justify their SB based on these distinctions. While knowledge is important, it is not sufficient by itself to effect lasting change. Additional strategies, such as individual goal setting and action planning, are needed to address established behaviors [46]. Different populations may need different strategies to reduce SB and a multifaceted approach is recommended.

Incorporating social interaction into programs for older adults is of paramount importance, as shown by some studies [42,46,47,48,56,60]. These studies show that older adults want programs that provide group activities and promote social interactions. Such programs not only help reduce sedentary activities, but also foster social relationships and combat feelings of isolation and loneliness. It is evident that enjoyable and engaging activities that promote social interactions are highly valued by older adults. In addition, staff support and encouragement prove to be critical factors in motivating residents of assisted living facilities to participate in these programs. Competent and dedicated staff who can guide residents and ensure their safety during exercise sessions are highly valued [60].

Program accessibility and convenience are critical factors influencing participation. Older adults prefer on-site programs that are easily accessible and meet their specific mobility needs. Flexible scheduling of activities based on individual preferences and daily routines is also important for engagement [60]. Convenience and simplicity are highlighted as important factors in activity design. Older people want individualized adaptations and options based on their preferences and activity levels. Key features include goal setting, notifications, self-monitoring, ease of use, and social support [56].

Older adults’ views on the use of technology in interventions to reduce SB have been explored in some studies that offer valuable insights into their preferences. Matson and colleagues [47] conducted a study that focused on obese older adults and their perceptions of a technology-based SB reduction intervention. Participants in this study expressed positive attitudes toward the intervention and considered it acceptable and easy to incorporate into their daily lives. A similar study by Trinh and colleagues [56] with prostate cancer survivors receiving androgen deprivation therapy found that some men were initially hesitant to use the SB technology for weight loss. However, after understanding that the intervention would be tailored to their needs and preferences, they expressed interest in trying it. Participants indicated a preference for wearable devices such as tablets or cell phones to deliver the intervention. In contrast, Leask and colleagues [45] found that participants preferred non-technological methods to change their SB. They cited previous negative experiences and concerns about discomfort and unwanted attention as reasons for their preference. Tangible approaches, such as keeping a diary to track activity, were preferred to technological solutions. These studies highlight older people’s differing views on the use of technology in SB reduction interventions. While some people are reluctant or prefer non-technological methods, others find technology-based interventions acceptable and easy to integrate into their lives. Therefore, it is important to consider individual preferences when developing interventions to reduce SB in older adults.

The findings of our review are consistent with the social-ecological model [76] and underscore the complex nature of this behavior. Specifically, SB in older adults is influenced by multiple factors at the individual, social, and environmental levels. When developing strategies, it is important to consider these interrelated factors and ensure that interventions can be seamlessly integrated into older adults’ daily lives with the clear goal of effectively reducing SB. By implementing such strategies, we can make significant progress in addressing persistent SB in older people. When developing SB interventions, it is also important to consider not only behavior change techniques, but also how they are implemented. The behavior change wheel [86] emphasizes the importance of factors such as enjoyment, social interaction, and convenience, especially for older adults. Individual SB reduction goals should be established to guide the selection and intensity of behavior change strategies. This approach involves understanding the beliefs underlying SB and adapting alternatives to existing SB. When these considerations are considered, interventions can be better tailored to the individual and increase their effectiveness in promoting behavior change.

In summary, interventions to reduce SB in older adults should emphasize individual adaptation, social interaction, choice, and education. By incorporating these elements, programs can effectively address the individual needs and preferences of older adults, promote active aging, enhance well-being, and empower individuals to adopt and maintain healthier lifestyles. A person-centered, inclusive approach to developing and implementing programs to reduce SB is critical to the success and long-term impact of these interventions.

## 5. Insights from Qualitative Research on Sedentary Behavior: A Conclusion

This scoping review of qualitative studies has identified five important research trends that offer valuable insights into understanding SB among older adults. First, the definition and shaping perspectives of SB demonstrate the importance of clarifying the concept and exploring individual perceptions of SB. Second, the dynamics of daily routines and contexts underscore the contextual factors that influence SB, such as work, leisure activities, and social environment. The third theme, perceived benefits and drawbacks, highlights the perceived advantages and disadvantages associated with SB and emphasizes the need to increase awareness and understanding of its impact on health and well-being. The fourth theme, determinants and discouraging factors, addresses the factors and barriers that contribute to a sedentary lifestyle, including individual, social, environmental, and cultural factors. Finally, the theme related to interventions addresses older adults’ views of the barriers and motivators to participating in SB reduction programs, the perceived effects of participating in such programs, and strategies suggested by older adults for implementing these types of interventions. This theme highlights the importance of implementing evidence-based interventions that address multiple levels, including older adults’ perspectives. Another finding is that, despite the diverse backgrounds of the subjects—from healthy to obese—and their different living circumstances, whether in a community or in an institution, a comprehensive qualitative examination of views on SB consistently yields analogous assessments on various aspects across cohorts. These consistent perceptions underscore that SB is widespread across populations and transcends linguistic differences between Anglo-Saxon and non-Anglo-Saxon populations.

Figure 2 provides a comprehensive visual representation of the five major themes that emerged from the qualitative research, offering valuable insights into the complex relationship between older adults and SB.

Taken together, these themes provide a comprehensive understanding of SB and its complexity. They highlight the multifaceted nature of SB and the need for a holistic approach to solving the problem. By considering these themes, researchers, practitioners, and policymakers can develop tailored interventions and strategies to promote active lifestyles and reduce SB in older adults. Further qualitative research is warranted to delve deeper into these themes and explore additional nuances and perspectives that can contribute to a thorough understanding of SB and enable effective interventions and strategies. As Lewin and Glenton [87] point out, qualitative research supports the development of World Health Organization strategies and interventions through its benefits. These benefits include a deeper understanding of complex phenomena, exploration of contextual insights that influence health-related behaviors and interventions, involvement of stakeholders in the research process, use of an iterative approach to continuous improvement, and complementing quantitative data with deeper insights and explanations.

Analysis of qualitative research provides valuable practical implications and considerations for promoting active aging and reducing SB in older adults. It is critical to tailor interventions to individual needs and abilities, considering preferences, age, and physical abilities. Strategies should focus on increasing motivation and confidence. Incorporating available resources such as information, social support, and physical and psychological resources into daily activities can significantly reduce SB. Educating older adults about the health benefits of reducing SB and providing practical advice on how to incorporate more physical activity into their daily lives are critical to increasing knowledge and motivation. In addition, social support and engagement can help combat loneliness and promote active lifestyles. Creating easily accessible local activities that consider the unique mobility needs, preferences, and daily routines of older people encourages engagement. It is also important to consider older people’s different views on the use of technology and incorporate appropriate technological or non-technological solutions based on their preferences. By implementing these meaningful technologies and addressing these concerns in combination with evidence-based human movement behaviors [88], we can successfully minimize SB, promote active aging, and improve the overall well-being and quality of life of older adults.

Although this scoping review of qualitative studies provides valuable insights into SB among older adults, it is important to acknowledge its limitations. First, the inclusion of only qualitative studies may limit the generalizability of the findings because qualitative research focuses on in-depth investigation rather than statistical representation. In addition, the selected studies were limited to a specific time period, language, and predominantly Western context, which limits the transferability of findings to other regions. To gain a more comprehensive understanding, it is important to examine the experiences of SB in different cultures and populations. By examining a broader range of contexts, we can gain additional insights into the complicated interplay of cultural norms, societal factors, and SB. Despite these limitations, this review provides insights into qualitative research on SB among older adults and offers a foundation for future research efforts in this area.

To further advance the field, future studies may benefit from including quantitative measures in addition to qualitative outcomes to improve the comprehensive understanding of SB in older adults. While this review focused primarily on the perspectives of older adults themselves, it would be valuable to also examine the perspectives of health professionals, caregivers, and other stakeholders involved in promoting active lifestyles in this population. In addition, future qualitative studies can explore targeted strategies and interventions that effectively address barriers to reducing SB among older adults. By examining the experiences of older adults in different contexts, such as transitioning from work to retirement or living in different environments, valuable insights can be gained to tailor interventions to specific circumstances. Likewise, examining the cultural and socioeconomic factors that influence SB among older adults from diverse backgrounds may lead to more comprehensive and targeted approaches. Further studies focusing on nursing home populations would deepen our understanding of interventions to reduce SB in these settings. Research on older adults in rural settings and consideration of the setting and its impact on SB could also provide insights into the unique challenges and opportunities faced by older adults in such settings, including factors such as access to resources, social support, and environmental characteristics.

Further qualitative research on specific mentally passive and active SB is also critical. While previous studies have provided significant insights into SB, there is a need to explore the nuances and complexities of these behaviors more deeply. Qualitative research allows for a more in-depth examination of the experiences, motivations, and contextual factors associated with SB. It can help uncover the deeper reasons for passive or active SB, shed light on social and environmental influences, and provide a deeper understanding of the potential barriers to and facilitators of behavior change.

Further considerations should be considered in future studies to improve our understanding of SB in older adults. Ethnographic approaches that capture the context and experiences of older adults may provide valuable insights into the unique barriers, facilitators, perceptions, and needs associated with reducing SB. In addition, an ecological momentary assessment combined with interviews or focus groups would provide a deeper understanding of the moment-to-moment experiences and contextual factors that influence SB. This real-time data collection approach would shed light on the situational determinants and motivations for SB among older adults. Examining factors that influence SB, such as the effectiveness of prompts, reminders, or environmental changes, would also provide valuable insights for developing behavior change interventions. Evaluating the impact of SB interventions, including their effectiveness, sustainability, and long-term outcomes, can inform evidence-based practices to promote active lifestyles among older adults. Finally, there is a need to further explore older adults’ perspectives on integrating technology into SB interventions to reduce physical inactivity. By addressing these areas, future studies can expand our knowledge and develop strategies to effectively address SB in the older population.

Overall, this review of qualitative studies contributes to our understanding of the complexity of SB in older adults and highlights the need for multifaceted and tailored interventions that consider individual, social, and environmental factors. By considering these findings, future interventions and strategies can effectively promote active aging and reduce SB, ultimately improving the overall well-being and quality of life of older people.

## Figures and Tables

**Figure 1 healthcare-11-02215-f001:**
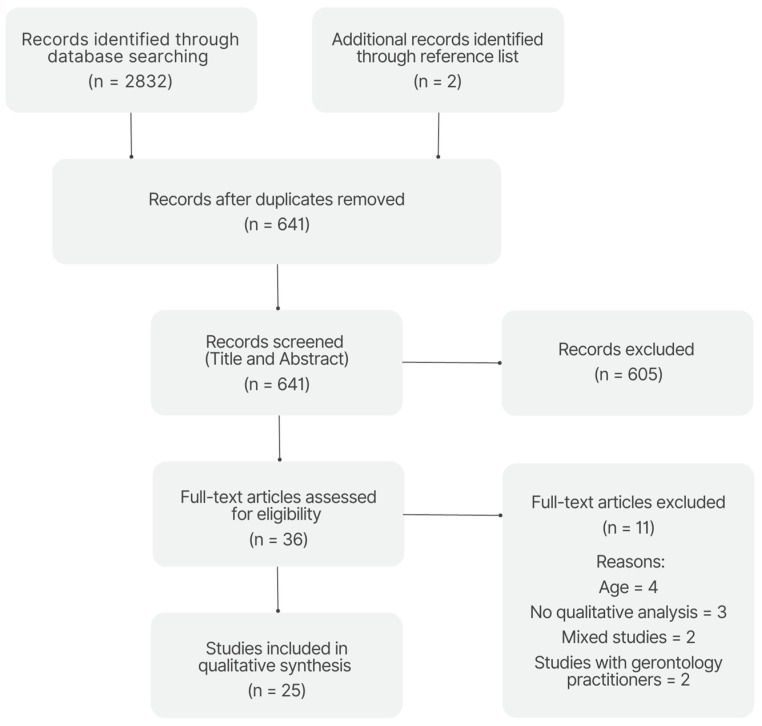
Results from the literature search.

**Figure 2 healthcare-11-02215-f002:**
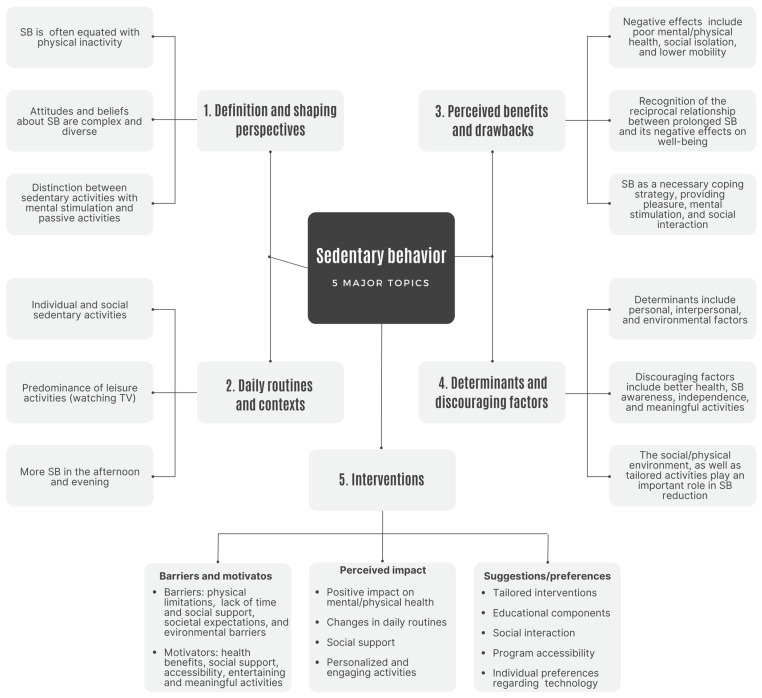
Summary diagram on the five major themes that emerged from research.

**Table 1 healthcare-11-02215-t001:** Search strategy in scientific databases.

1. Sedentary lifestyle [MeSH Terms] 2. (((sedentary behavior [tiab]) OR sedentary behaviour [tiab]) OR sedentary lifestyle [tiab]) 3. prolonged sitting [tiab] 4. Computer [MeSH Terms] 5. ((computer use [tiab]) OR computer usage [tiab]) OR computer time [tiab] 6. Television [MeSH Terms] 7. ((television viewing [tiab]) OR television watching [tiab]) OR television time [tiab] 8. ((TV viewing [tiab]) OR TV watching [tiab]) OR TV time [tiab] 9. ((screen watching [tiab]) OR screen use [tiab]) OR screen time [tiab] 10. ((screen entertainment [tiab]) OR screen behaviour [tiab]) OR screen behavior [tiab] 11. reading time [tiab] 12. automobile driving [MeSH Terms] 13. transport time [tiab]
AND
14. qualitative research [MeSH Terms] 15. ethnography [tiab] 16. phenomenology [tiab] 17. life stories [tiab] 18. grounded theory [tiab] 19. case studies [tiab] 20. focus groups [tiab] 21. descriptive qualitative study [tiab] 22. qualitative investigation [tiab] 23. qualitative analysis [MeSH Terms] 24. thematic analysis [tiab] 25. content analysis [tiab] 26. interpretive phenomenological analysis [tiab] 27. thematic synthesis [tiab] 28. narrative analysis [tiab] 29. interviews [MeSH Terms] 30. in-depth interviews [MeSH Terms] 31. structured interviews [tiab] 32. semi-structured interviews [tiab] 33. unstructured interviews [tiab] 34. ((observation [tiab]) OR participant observation [tiab])
AND
35. aging [MeSH Terms] 36. (senior [tiab]) OR senior citizens [tiab] 37. older adult [tiab] 38. elderly [tiab] 39. older people [tiab] 40. geriatric OR geriatrics

**Table 2 healthcare-11-02215-t002:** Descriptive characteristics of the studies.

Author (Year/Country)	Research Purposes	Study Population	Data Collection	Data Analysis	Major Themes
Biswas et al. (2018) Canada [37]	To explore the level of awareness among cardiac rehabilitation patients and the staff regarding SB, as well as the perceived facilitators and barriers to reducing SB	Cardiac rehabilitation patients *n* = 15 (47% male) 63.00 (±10.6)	Semi-structured interview	Thematic analysis	Benefits/Drawbacks Determinants/Discouragers
Brookfield et al. (2015) UK [38]	To examine older adults’ experiences of their home environment and its influence on their physical activity and SB	Older adults: healthy volunteers stroke survivors, and people with dementia *n* = 22 (27% male)	Semi-structured interview and focus group	Inductive thematic analysis	Daily routines/Contexts
Chastin et al. (2014) UK [39]	To capture older adults’ perspectives and opinions on the factors that influence their sedentary behavior	Healthy, community-dwelling older women *n* = 9 (0% male) 79.44 (±7.75)	Semi-structured interview	Framework analysis and inductive thematic analysis	Definition/Perspectives Daily routines/Contexts Benefits/Drawbacks Determinants/Discouragers
Collins & Pope (2021) USA [40]	To examine the degree to which older adults internalize their motives for limiting SB, consistent with self-determination theory	Community-dwelling older adults *n* = 27	Focus group	Thematic analysis	Determinants/Discouragers
Eklund et al. (2021) Sweden [41]	To gain a comprehensive understanding of SB as it relates to the transition from work to retirement, specifically how it is experienced by individuals in retirement	Retired older adults *n* = 14 (43% male) 70.36	Semi-structured interview	Empirical phenomenological psychological method	Definition/Perspectives Benefits/Drawbacks Determinants/Discouragers
Gour et al. (2020) India [42]	To explore older adults’ views on the usefulness of yoga or light exercise and its effects on well-being and the prevention of SB	Older people who participated in an RCT intervention 66.00	Focus group based on a semi-structured guide	Inductive content analysis	Interventions
Greenwood-Hickman et al. (2016) USA [43]	To examine the motivators, barriers, and effects of SB on a group of overweight and obese older adults	Overweight and obese older adults following a SB reduction intervention *n* = 24 (33% male) 72.00	Semi-structured interview	Inductive thematic analysis	Determinants/Discouragers Interventions
Kotlarczyk et al. (2020) USA [44]	To examine the factors contributing to SB among residents of both independent and assisted living facilities	Older residents of independent and assisted living facilities *n* = 44 (30% male) 86.00	Focus group	Thematic analysis	Definition/Perspectives Benefits/Drawbacks Determinants/Discouragers
Leask et al. (2016) UK [45]	To understand older adults’ perspectives and identify critical factors to consider when developing a solution to change daily sedentary patterns	Healthy, community-dwelling older adults *n* = 15 (27% male) 78.00	Focus group based on semi-structured interview	Thematic analysis	Interventions
Leask et al. (2017) UK [46]	To co-create a tailored public health intervention to reduce SB in older adults	Healthy community-dwelling older adults *n* = 11 (46% male) 74.00 (±5.5)	Semi-structured workshop	Content analysis	Interventions
Matson et al. (2018) USA [47]	To explore older adults’ acceptance of a theory-based and technology-enhanced sitting reduction intervention	Obese, community-dwelling older adults *n* = 22 (36% male) 69.20 (±4.9)	Semi-structured interview	Inductive and deductive thematic analysis	Interventions
McCain et al. (2023) USA [48]	To explore the acceptability of a standing intervention among older adults living in long-term care with the guidance of the Theoretical Framework of Acceptability	Older adults living in long-term care from the intervention trial *n* = 10 (30% male) age 73 to 102 years	Semi-structured interview	Inductive and deductive thematic framework analysis	Interventions
McEwan et al. (2017) Canada [49]	To better understand older adults’ perceptions of SB, its pros and cons, and the barriers associated with reducing sedentary time	Healthy, community-dwelling older adults *n* = 25 (33% male) 74.00 (±8.5)	Focus group based on a semi-structured interview	Directed content analysis	Definition/Perspectives Benefits/Drawbacks Determinants/Discouragers
McGowan et al. (2019) UK [50]	To explore the factors influencing SB in older adults and to evaluate the acceptability of potential strategies to reduce SB	Healthy, community-dwelling older adults *n* = 22 (36% male) 76.00	Semi-structured interview	Inductive thematic analysis	Definition/Perspectives Benefits/Drawbacks Determinants/Discouragers
McGowan et al. (2021) UK [9]	To examine older adults’ understanding of the concept of SB	Community-dwelling older adults *n* = 22 77.50	Semi-structured interview	Inductive thematic analysis	Definition/Perspectives Benefits/Drawbacks Determinants/Discouragers
Moraes et al. (2020) Brazil [51]	To understand older adults’ experiences after hip fracture surgery considering barriers and facilitators related to SB	Older adults who had hip fracture surgery *n* = 11 (27% male)	Semi-structured interview	Inductive thematic analysis	Determinants/Discouragers
Nuwere et al. (2022) USA [52]	To understand older adults’ perceptions regarding SB and its influence on healthy aging	Community-dwelling older adults who attended two senior centers *n* = 46 (11% male) 75.60 (± 7.8)	Focus group based on a semi-structured discussion guide	Thematic analysis	Definition/Perspectives Daily routines/Contexts Benefits/Drawbacks Determinants/Discouragers
Palmer et al. (2019) UK [53]	Compare the types, context, and role of sitting activities in the daily lives of older adults who sit more or less than average	Healthy, community-dwelling older adults *n* = 44 (52% male) 74.60	Semi-structured interview	Inductive thematic analysis	Definition/Perspectives Daily routines/Contexts Determinants/Discouragers
Palmer et al. (2021) UK [54]	To investigate how older adults make sense of their SB	Older adults who participated in a previous project *n* = 44 (52% male) Not reported	Semi-structured interview	Thematic analysis	Definition/Perspectives Daily routines/Contexts Benefits/Drawbacks Determinants/Discouragers
Tam-Seto et al. (2016) Canada [55]	To use data on perceptions of sedentary time and the programs/supports that older adults perceive as important to reducing their sedentary time	Healthy, community-dwelling older adults *n* = 26 (23% male) 74.00 (±8.5)	Focus group based on semi-structured interview	Directed content analysis	Determinants/Discouragers
Trinh et al. (2015) Canada [56]	To describe and understand the perceptions of SB and interests and preferences for a SB intervention in men undergoing androgen deprivation therapy	Male prostate cancer survivors *n* = 27 (100% male) 73.50 (±8.1)	Focus group interview	Deductive thematic synthesis	Definition/Perspectives Benefits/Drawbacks Interventions
Van Dyck et al. (2017) Belgium [57]	To obtain qualitative information on change and the specific multidimensional determinants of physical activity and SB during early retirement, on recently retired adults’ opinions of existing PA programs/interventions, and on their needs and desires for new interventions	Healthy, community-dwelling older adults *n* = 37 (49% male) 62.90 (±1.9);	Focus group based on semi-structured interview	Inductive and deductive thematic analysis	Definition/Perspectives Daily routines/Contexts Determinants/Discouragers
Voss et al. (2020) Canada [58]	To explore perceptions of sedentary time as well as barriers and motivators to reducing SB among older adults	Older adults in assisted living *n* = 31 (26% male) 83.50 (±6.5)	Semi-structured focus group	Thematic analysis	Definition/Perspectives Benefits/Drawbacks Determinants/Discouragers
Webber et al. (2020) Canada [59]	To explore the understanding of physical activity and SB, including barriers and health influences in people with osteoarthritis and knee arthroplasty	Older adults with osteoarthritis and knee arthroplasty *n* = 22 (36% male) 67.50 (±5.3) (pre-operative) and 67.3 (±7.0) (post-operative)	Focus group interview	Inductive thematic analysis	Definition/Perspectives Benefits/Drawbacks Determinants/Discouragers
Webster et al. (2023) USA [60]	To collect older adults’ in assisted living recommendations on a proposed intervention to increase self-efficacy to replace SB with light physical activity	Older adults in assisted living *n* = 20 (40% male) 83.1	Interview	Content and thematic analysis	Benefits/Drawbacks Determinants/Discouragers Interventions

## Data Availability

Not applicable.

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
