# Peer review of "The Quiet Epidemic: An Overview of Emerging Qualitative Research Trends on Sedentary Behavior in Aging Populations"

_healthcare, 2023, doi:10.3390/healthcare11152215_

Round 1

Reviewer 1 Report

It's an interesting study with captions (headings) that capture readers' attention. It's going to contribute greatly to the field. There are minor suggestions for consideration.

1. Breaking Ground on Sedentary Behavior: An Introduction to a Silent Threat

As several reviews on SB among older adults have been conducted, it should emphasize at the beginning of the study why including only qualitative research is needed to give magnitude and significance of the review.

2. Crafting the Map: Eligibility Criteria and Search Strategy

- As SPIDER is used to provide inclusion criteria, it would be great if each alphabet of the acronym is presented in order. It would ease readers to follow along and make sure that all criteria are met.

- Search strategies including keywords for each database should be included as an additional file. This would show transparency and reducibility of the review and other researchers can learn from the review.

3. From Palette to Canvas: Study Characteristics

As a review aims to gather as many studies as possible to pool data and provide conclusion, the number of studies reviewed and included in the review is expected. A total number of studies included in the review should be reported at the beginning of the session. A flowchart reporting how many articles retrieved from each database (3 databases and reference lists) and each step (showing how collected articles were selected). This would increase reliability of the review.

4. Cracking the Code of Sedentary Behavior: Key Themes Explored and Discussed

On page 10, please removing repeating phrase 'The findings of'

Author Response

Dear esteemed reviewer,

We are very pleased to receive your valuable comments on our scholarly article. Your thoughtful feedback is truly invaluable and we believe it will play a critical role in improving the quality and impact of our research.

The comments received and the actions we have taken are attached at the end of this letter (Please see the attachment). We have also attached an updated version of our manuscript with all changes highlighted in blue.

Once again, we are very grateful for the time and effort you took to review our work.

Best regards,

Reviewer 2 Report

This is a comprehensive review of qualitative research on sedentary behaviour for older adults and makes many interesting points. My comments are below:

Methods:

1) The methodology used appears to be that of a scoping review rather than a narrative review. Developing a search strategy and conducting a search and extracting and analysing data aligns with scoping review methodology. The aim of identifying knowledge gaps and areas for future research also aligns with scoping review aims. This is a major flaw in this article. This should be changed to describe the article as a scoping review. See Munn Z, Peters MD, Stern C, Tufanaru C, McArthur A, Aromataris E. Systematic review or scoping review? Guidance for authors when choosing between a systematic or scoping review approach. BMC medical research methodology. 2018 Dec;18:1-7. 

2) Section 2, titled crafting the map: eligibility criteria and search strategy - This section also describes data extraction and analysis/synthesis methods, and the title should be amended to show that. 

3) I would like to see an example of the search strategy included in the paper or supplementary material. 

4) The manuscript lacks any detail about the study screening and selection process for determining which studies should be included. This is essential for reviews. 

5) Line 123: This line states "the included studies were systematically analysed (data extraction) for their characteristics" - data extraction refers to collecting data, not analysing it, so this is confusing. This should be amended for accuracy. 

Results (section 3): 

1) Line 146: This should say 'focus groups' instead of 'focus group interviews'

2) There is no detail on how many articles the search returned and no breakdown of the screening process, such as how many articles were removed after title and abstract screening, and then full text screening. A flow diagram would be helpful here to show this. 

3) Line 262: There is repetition of 'The findings'. 

Author Response

(The authors gave the same response as above.)

Reviewer 3 Report

Altogether interesting and appropriate, perhaps slightly verbose in outline. I believe it could be further made interesting by adding a comment on the presence or absence of differences between the "healthy" population and the obese or post interventions. Similarly, despite the numerical limitations, it could be useful to comment on any quantitative differences in the interpretations between Anglo-Saxon and non-Anglo-Saxon populations.

Author Response

(The authors gave the same response as above.)

Round 2

Reviewer 2 Report

Thank you for responding to my initial comments. I believe you have responded to them adequately.

My only remaining comments are regarding the abstract where it still refers to the study as a narrative review - this should be changed to scoping review. It should also say how many articles were included in the review. 

Author Response

Dear esteemed reviewer,

Thank you again for your insightful and precise comments. 

The actions we have taken are attached in an updated version of our manuscript with all changes highlighted in blue (abstract).

Once again, we are very grateful for the time and effort you took to review our work.

Best regards,